# LightRefine-PCXR: A Lightweight Refinement Framework for Efficient Medical Device Suppression in Pediatric Chest X-Rays

**Mingze Jiang**[1]                                                    MJIANG23@ND.EDU
**Xueyang Li**[1]                                                          XLI34@ND.EDU
**John Kheir**[2,3]                                   JOHN.KHEIR@CARDIO.CHBOSTON.ORG
**Alec Girten**[3]                               ALEC.GIRTEN@CHILDRENS.HARVARD.EDU
**Yiyu Shi**[1]                                                             YSHI4@ND.EDU

[1] *University of Notre Dame, Notre Dame, USA*

[2] *Harvard Medical School, Boston, USA*

[3] *Boston Children's Hospital, Boston, USA*

**Editors:** Accepted for publication at MIDL 2026

## Abstract

In pediatric chest radiography, indwelling support devices (e.g., tubes and lines) are ubiquitous and often obscure critical thoracic structures, complicating radiologic interpretation and reducing the reliability of automated analysis methods. Although generative inpainting has advanced rapidly, reliable deployment in pediatric chest radiographs remains challenging. Subtle anatomical cues must be preserved under substantial domain shift, while full adaptation of large backbones is often impractical because of limited pediatric data and constrained clinical compute budgets. To address these limitations, we propose LightRefine-PCXR, a lightweight, backbone-agnostic refinement framework for suppressing medical devices in pediatric chest X-rays (PCXRs). LightRefine-PCXR follows a two-stage strategy: a frozen pretrained inpainting backbone first produces a coarse device-removed estimate, and a compact anatomy-aware refiner then predicts mask-constrained residual corrections to restore local structures while preserving all unmasked pixels exactly. This plug-in design substantially reduces trainable parameters and peak GPU memory compared with end-to-end fine-tuning, yet consistently improves reconstruction fidelity and perceptual quality across diverse inpainting paradigms, including CNN-, transformer-, and diffusion-based models. Comprehensive in-domain and cross-dataset experiments demonstrate robust device suppression and strong generalization in low-data pediatric settings, highlighting the practicality of LightRefine-PCXR for real-world pediatric radiology workflows. Our code is available at https://github.com/jiangmingze42/LightRefine-PCXR.

**Keywords:** Medical Device Suppression, Pediatric Chest X-ray, Resource-efficient Refinement, Plug-in Module, Generative Inpainting

## 1. Introduction

Pediatric chest X-rays (PCXRs) are among the most critical radiographic examinations in pediatric emergency and intensive care settings, supporting the diagnosis of cardiopulmonary diseases, assessment of treatment response, and monitoring of mechanically ventilated patients (Padash et al., 2022). However, PCXRs frequently include medical devices, such as endotracheal tubes, nasogastric tubes, and central venous lines, which often

project over anatomically important regions. These high-contrast structures can obscure radiographic findings, hinder the recognition of subtle disease patterns, and degrade the reliability of downstream automated analysis systems (Seah et al., 2021; Tang et al., 2021). Accordingly, as clinical workflows increasingly incorporate AI-assisted decision-making (Li et al., 2023), anatomically faithful medical device suppression has become important for improving diagnostic clarity and supporting robust downstream model performance (Çallı et al., 2021).

Despite clear clinical demand, reliable device suppression in pediatric chest X-rays remains insufficiently explored (Padash et al., 2022). Existing approaches designed for adults often fail to transfer effectively, as pediatric radiographs possess distinct anatomical and developmental characteristics, and many available pipelines are either closed-source or heavily optimized for adult imaging (Pedrosa et al., 2024; Jin et al., 2024; Li et al., 2025). In addition to these transferability challenges, global generative reconstruction introduces additional safety concerns, as unconstrained modifications may generate spurious structures or obscure pathologies (Kumar et al., 2025; Bhadra et al., 2021). Furthermore, inpainting models pretrained on natural images (Suvorov et al., 2021; Li et al., 2022) are particularly vulnerable to pediatric domain shifts, potentially compromising subtle radiographic cues unless specifically adapted. Although full fine-tuning of large generative models can alleviate these issues, such training is often impractical in clinical environments due to limitations in pediatric data availability (Pham et al., 2023; Kermany et al., 2018), as well as significant computational and memory demands (Linguraru et al., 2024), and the risk of overfitting to small datasets. Recent advances in computer vision have proposed lightweight adaptation strategies, such as conditional controllers and adapter-based modules (Zhang et al., 2023; Mou et al., 2024); however, these techniques have primarily been evaluated on natural images, with limited evidence supporting their reliability or safety in medical radiography.

In light of these limitations, we present LightRefine-PCXR, a refinement-based framework that makes pediatric device suppression both efficient and anatomically faithful. Rather than adapting large generators end to end, LightRefine-PCXR decouples reconstruction into two complementary stages: a frozen, pretrained inpainting backbone first provides a coarse device-removed estimate, and a compact refiner then learns mask-restricted residual updates that correct local structural errors while exactly preserving all unmasked pixels. This plug-in design is inherently backbone-agnostic, allowing seamless integration with CNN-, transformer-, and diffusion-based inpainting models and enabling pediatric adaptation under limited data and compute. Across VinDr-PCXR (Pham et al., 2022) and external testing on CHD-CXR (Zhixin et al., 2024), our model consistently strengthens reconstruction fidelity and perceptual quality relative to backbone-only inference, while markedly reducing trainable parameters and peak training memory compared with full fine-tuning. Together, these results position LightRefine-PCXR as a deployable and safety-conscious solution for medical device suppression in resource-constrained pediatric radiology workflows.

## 2. Methodology

### 2.1. Overview

Fig. 1 illustrates LightRefine-PCXR, a two-stage framework for pediatric medical device suppression that combines coarse inpainting with fine inpainting. In the first stage, a

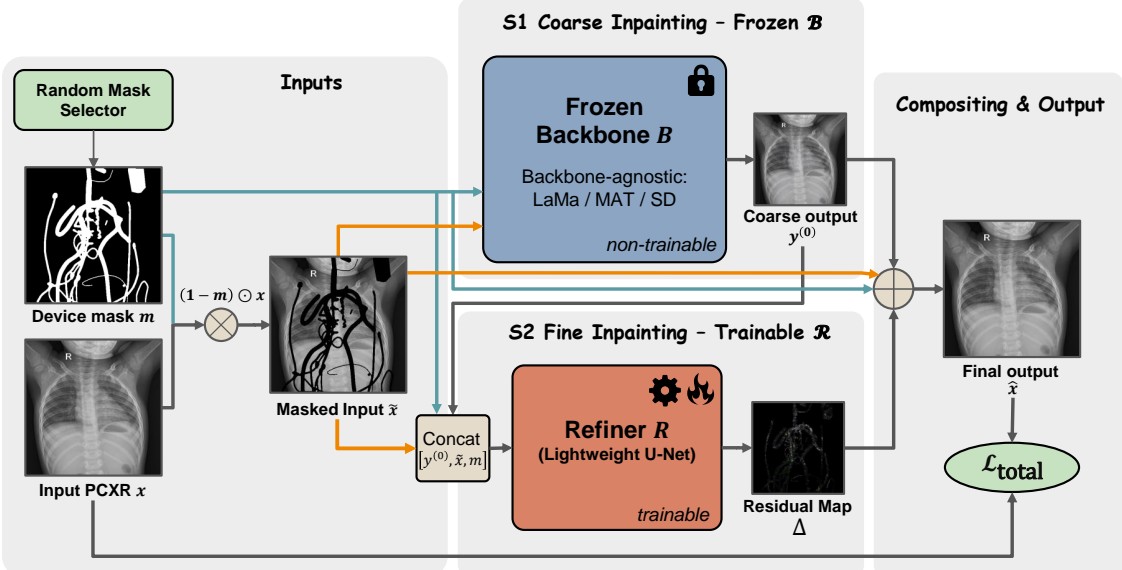

Figure 1: **Overview of LightRefine-PCXR.** *Left (train/val):* a clinically annotated device mask is sampled from the mask pool and composed with a ground-truth device-free pediatric chest X-ray $x$ to form the masked input $\tilde{x}$. *Middle (Two-Stage Refinement):* **S1** a frozen pretrained inpainting backbone $B$ generates a coarse completion $y^{(0)}$; **S2** a lightweight U-Net (Ronneberger et al., 2015) refiner $R$ predicts a residual $\Delta$ applied only within the device mask. *Right:* the final output $\hat{x}$ preserves all unmasked anatomy exactly by construction. At deployment, the same pipeline is applied to device-present CXRs using masks from manual or automatic device segmentation. The framework is backbone-agnostic and supports CNN-, transformer-, and diffusion-based inpainting models.

frozen pretrained inpainting backbone produces a coarse device-removed estimate from the masked input, leveraging strong generic priors without pediatric-specific fine-tuning. In the second stage, a compact refinement module performs fine inpainting by predicting mask-constrained residual corrections, targeting local anatomical artifacts while preserving all unmasked pixels exactly by construction. This plug-in formulation decouples pediatric adaptation from the large generator, enabling resource-efficient training under limited data and compute, and it remains compatible with diverse inpainting paradigms. The backbone, refiner, and loss functions are detailed in the below sections.

**Training vs. deployment protocol.** LightRefine-PCXR is trained as a masked completion problem and does not require paired device-present and device-free CXRs: during training and validation, we use device-free pediatric CXRs as reconstruction targets and sample clinically annotated device masks from an independent mask pool to define the edit region. The same setting is used for held-out validation and test-time evaluation with synthetic masks. In all cases, the backbone and refiner operate on the masked in-

put (masked pixels are removed and never observed). At deployment, however, the source CXR is device-present and masks come from manual annotation or an automatic device segmentation module, while the same two-stage refinement pipeline is applied.

## 2.2. Backbone for Coarse Inpainting

We employ a pretrained inpainting backbone to produce an initial device-suppressed estimate. The backbone is kept fixed to retain its pretrained generative prior and to avoid the cost and potential instability of end-to-end adaptation under pediatric domain shift.

Let $x \in \mathbb{R}^{3 \times H \times W}$ denote a pediatric CXR, formed by replicating the original grayscale image into three channels to match the input interface of common pretrained inpainting models and perceptual metrics. Let $m \in \{0, 1\}^{1 \times H \times W}$ denote a binary device mask.

We mask the device region to form $\tilde{x}$ and feed the masked image and mask into the backbone $B$ to obtain a coarse completion:

$$\tilde{x} = (1 - m) \odot x, \qquad y^{(0)} = B([\tilde{x}, m]). \tag{1}$$

By freezing $B$, pediatric adaptation is delegated to the lightweight refiner, which reduces training memory and computation and enables plug-and-play replacement across different inpainting families.

## 2.3. Mask-constrained Residual Refiner for Fine Inpainting

The fine inpainting stage is realized by a lightweight residual refiner that operates exclusively within the device mask and corrects pediatric-specific artifacts in the frozen backbone output. Confining all learned updates to the masked region imposes a hard guarantee that unmasked anatomy is preserved exactly.

The refiner $R$ is a compact U-Net (Ronneberger et al., 2015) with a ResNet-34 (He et al., 2016) encoder and a symmetric decoder with skip connections. It takes as input the backbone prediction, the masked image, and the mask, i.e., $[y^{(0)}, \tilde{x}, m] \in \mathbb{R}^{7 \times H \times W}$, and predicts a three-channel residual:

$$\Delta = R\Big([y^{(0)}, \tilde{x}, m]\Big), \qquad \Delta \in \mathbb{R}^{3 \times H \times W}. \tag{2}$$

The residual formulation encourages $R$ to concentrate on localized corrections inside the mask (e.g., edge continuity and intensity consistency) rather than re-synthesizing the full image. The residual is applied only within the mask and the final output is composed as

$$y^{(1)} = y^{(0)} + m \odot \Delta, \qquad \hat{x} = (1 - m) \odot x + m \odot y^{(1)}. \tag{3}$$

## 2.4. Anatomy-preserving Loss Functions

To complement the mask-constrained refinement design, we adopt anatomy-preserving loss functions that explicitly prioritize faithful reconstruction within the device region while maintaining smooth transitions at its boundary. The objective couples residual supervision with structure-oriented regularizers, which collectively encourage edge continuity and local perceptual realism in the edited area without incentivizing changes to unmasked pixels.

During training, given the ground-truth device-free image $x$, we define the target residual within the device mask as

$$r^* = m \odot \left(x - y^{(0)}\right), \tag{4}$$

which specifies the desired correction only in the device region.

**Masked residual supervision.** The primary term directly supervises the predicted residual inside the mask:

$$\mathcal{L}_{\text{res}} = \left\| m \odot \Delta - r^* \right\|_1. \tag{5}$$

**Structural regularization.** To encourage anatomical continuity and seamless blending, we add three complementary regularizers: an edge-consistency term based on Sobel gradients, a localized total-variation (Rudin et al., 1992) term applied over the mask and a narrow boundary band, and a masked perceptual term:

$$\begin{aligned}
\mathcal{L}_{\text{total}} = {} & \lambda_{\text{res}}\, \mathcal{L}_{\text{res}} + \lambda_{\text{edge}} \left\| m \odot \left(G(\hat{x}) - G(x)\right) \right\|_1 \\
& + \lambda_{\text{tv}}\, \text{TV}(\hat{x};\, m \cup \mathcal{B}(m)) + \lambda_{\text{lpips}}\, \text{LPIPS}(\hat{x}, x;\, m),
\end{aligned} \tag{6}$$

where $G(\cdot)$ denotes Sobel gradients computed on the grayscale intensity (converted from the three-channel representation), and LPIPS (Zhang et al., 2018) is computed only within $m$. $\mathcal{B}(m)$ denotes a narrow boundary band around the mask to promote smooth transitions across the device boundary. Together, these loss functions preserve fine anatomical edges, reduce local artifacts, and improve texture consistency within the masked region, while leaving unmasked anatomy unchanged by design.

In summary, LightRefine-PCXR integrates coarse inpainting from a frozen pretrained backbone with fine inpainting from a compact mask-constrained residual refiner. By restricting learned corrections to the device region and enforcing exact copying of unmasked pixels, the framework improves local anatomical fidelity and perceptual quality while enabling resource-efficient pediatric adaptation without end-to-end backbone fine-tuning.

## 3. Experiments

### 3.1. Datasets and Splits

We conduct experiments on two pediatric chest X-ray datasets: VinDr-PCXR (PediCXR) (Pham et al., 2022, 2023) and CHD-CXR (Zhixin et al., 2024). All models are trained on VinDr-PCXR only. We report in-domain performance on the official VinDr-PCXR test split and evaluate cross-dataset generalization on the full CHD-CXR dataset, which is used purely as an external test set without any fine-tuning.

In addition to these public datasets, we use an internal pediatric intensive care cohort to provide clinically realistic device masks. This cohort contains 1,000 clinically acquired PCXRs with manually annotated medical devices. Device trajectories were annotated by radiology technicians and verified by pediatric radiologists using polylines (e.g., endotracheal tubes, nasogastric tubes, and chest tubes) and polygons (e.g., vascular stents, permanent pacemaker markers, and ventilator tubing). After filtering empty or incomplete annotations, we obtain 942 non-empty device masks across 33 device categories, which are converted to binary masks and used as a mask pool for training (Fig. 2).

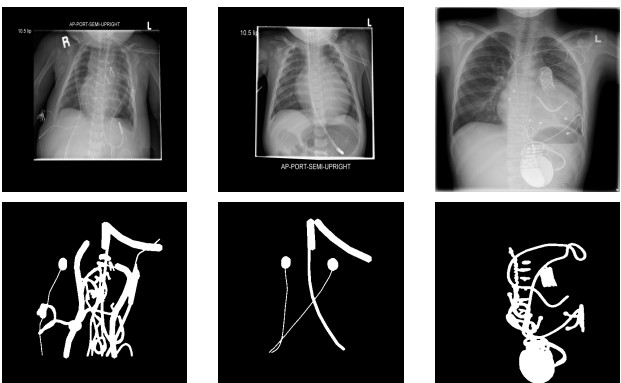

Figure 2: **Examples of device masks.** Top: original pediatric CXR images. Bottom: corresponding binary device masks derived from clinical annotations.

For VinDr-PCXR, we follow the official split, which provides only training and test partitions, and we further reserve 10% of the training partition as a validation set. Using the remaining training images, we construct four subsets containing 10%, 20%, 50%, and 100% of the data to study data efficiency. To prevent any data leakage, the device mask pool is split into disjoint training, validation, and test subsets that strictly mirror the image-level splits (test first, then 10% for validation from the remainder), and masks from different splits are never mixed. For each training fraction, we randomly sample the corresponding proportion of masks from the training mask pool (fixed after sampling) to ensure consistent supervision across experiments.

Image–mask pairing is then performed differently for training versus evaluation. During training, pairing is fully stochastic at the iteration level: at each training iteration, a device mask is independently and randomly sampled from the training mask pool and applied to the input image. As a result, the same pediatric CXR is paired with different masks across iterations, rather than with a fixed or predefined set of masks, which increases mask diversity per image and reduces overfitting to specific image–mask combinations. In contrast, during validation and testing, each image is paired with a fixed mask for reproducibility, and the same image–mask pairs are used consistently across all evaluated methods.

### 3.2. Experimental Setup

We instantiate LightRefine-PCXR with three representative inpainting backbones spanning different model families: LaMa (Suvorov et al., 2021) (**CNN-based**), MAT (Li et al., 2022) (**transformer-based**), and SD-v1.5 Inpaint (Rombach et al., 2022) (**diffusion-based**). All backbones are initialized from officially released pretrained checkpoints. Unless otherwise specified, backbone comparisons are conducted using the full VinDr-PCXR training portion after holding out the 10% validation split (i.e., the 100% setting in our data-fraction protocol). For each backbone, we compare **backbone-only inference** (no pediatric training) with **backbone + LightRefine**, where the backbone is kept frozen and only the refinement module is trained on VinDr-PCXR with learning rate $1 \times 10^{-4}$.

To quantify accuracy–efficiency trade-offs under a controlled training budget, we further focus on LaMa and run a head-to-head comparison between (1) **LaMa-SFT**, which fine-tunes the LaMa backbone end-to-end using the 100% training setting in our protocol, and (2) **LaMa + LightRefine**. In addition, we conduct a data-efficiency ablation by training the refiner with four data fractions (10%, 20%, 50%, and 100%) of the VinDr-PCXR training portion after holding out the validation split. All trainable LaMa-based configurations are optimized with the same learning rate ($1 \times 10^{-4}$) and the same fixed wall-clock budget of 12 hours to ensure a controlled comparison. LaMa-SFT uses the official default batch size of 15, whereas LightRefine uses batch size 32. For completeness, we also consider a low-rank PEFT baseline for LaMa based on DoRA-style weight decomposition (Liu et al., 2024) with adaptation applied to `Conv2d` layers, which is summarized alongside the efficiency analysis. We further evaluated ControlNet (Zhang et al., 2023) as a representative controllable-generation baseline; however, it performs poorly under our strict inpainting protocol and fails to provide the fidelity and spatial consistency required for clinical use (see Appendix C for details).

All CXRs and masks are preprocessed to $512 \times 512$ resolution via resizing or padding, and all experiments are benchmarked on a single NVIDIA A100 GPU (80 GB) for consistency, although LightRefine trains only a compact refinement module and can run on substantially smaller GPUs in practice. We report PSNR and SSIM (Wang et al., 2004) computed on full-image grayscale CXRs, and LPIPS (AlexNet) computed on RGB images following standard protocols. For the dedicated LaMa comparison, we additionally report trainable parameters and peak GPU memory usage.

## 4. Results and Discussions

### 4.1. Main comparison across backbones

Table 1 reports the main quantitative results on VinDr-PCXR (in-domain) and CHD-CXR (external). The **Backbone-only** setting applies the officially released pretrained checkpoints without pediatric training, whereas **Backbone + LightRefine** trains only the refinement module on the VinDr-PCXR training portion after holding out the validation split while keeping the backbone frozen. Across all three backbone families (LaMa, MAT, and SD-v1.5 Inpaint), LightRefine consistently improves PSNR and SSIM and reduces LPIPS on both datasets, while preserving unmasked anatomy exactly by construction. Notably, the improvements persist on CHD-CXR despite training being performed exclusively on VinDr-PCXR, suggesting that the refiner learns transferable, anatomy-consistent corrections rather than dataset-specific memorization.

The largest gains are observed for SD-v1.5 Inpaint, where LightRefine improves PSNR by 7.04 on VinDr-PCXR and 7.38 on CHD-CXR, substantially narrowing the gap to stronger deterministic inpainting backbones. Overall, the consistent improvements across CNN-, transformer-, and diffusion-based backbones indicate that the proposed residual refinement is broadly compatible with diverse generative priors.

Table 1: **Backbone-only vs. LightRefine.** We compare each pretrained inpainting backbone under (i) *Backbone-only* inference and (ii) *Backbone + LightRefine* (frozen backbone, trained refiner). Performance is evaluated using PSNR, SSIM, and LPIPS on VinDr-PCXR test (in-domain) and CHD-CXR (external). 'Improvement' denotes the absolute change of LightRefine over Backbone-only. ↑ indicates higher is better and ↓ indicates lower is better.

| Backbone | Method | VinDr-PCXR test | | | CHD-CXR | | |
|---|---|---|---|---|---|---|---|
| | | PSNR↑ | SSIM↑ | LPIPS↓ | PSNR↑ | SSIM↑ | LPIPS↓ |
| LaMa | Backbone-only | 41.23 | 0.9782 | 0.0124 | 40.39 | 0.9746 | 0.0126 |
| | w/ LightRefine | **43.73** | **0.9837** | **0.0068** | **42.14** | **0.9790** | **0.0083** |
| | Improvement | +2.50 | +0.0055 | ↓ 0.0056 | +1.75 | +0.0044 | ↓ 0.0043 |
| MAT | Backbone-only | 40.85 | 0.9761 | 0.0133 | 40.29 | 0.9730 | 0.0127 |
| | w/ LightRefine | **44.37** | **0.9862** | **0.0091** | **42.71** | **0.9819** | **0.0110** |
| | Improvement | +3.53 | +0.0101 | ↓ 0.0041 | +2.42 | +0.0090 | ↓ 0.0016 |
| SD-v1.5 Inpaint | Backbone-only | 33.22 | 0.9014 | 0.0385 | 31.80 | 0.8685 | 0.0376 |
| | w/ LightRefine | **40.27** | **0.9718** | **0.0140** | **39.18** | **0.9672** | **0.0153** |
| | Improvement | +7.04 | +0.0704 | ↓ 0.0245 | +7.38 | +0.0988 | ↓ 0.0223 |

## 4.2. Qualitative results and anatomical fidelity

Fig. 3 presents representative qualitative comparisons under two clinically relevant settings: (i) **device-present inputs**, which reflect the intended deployment scenario, and (ii) **device-free targets**, which serve as reconstruction supervision using synthetically masked inputs during training and evaluation. Across both settings, LightRefine yields visibly cleaner device suppression and more anatomically coherent reconstructions, with improved continuity of fine structures near the edited regions, thereby supporting reliable use in real-world device-present examinations.

For LaMa (top), the backbone output often leaves a conspicuous gray residual trace along the masked tube trajectory, which disrupts local tissue continuity and can obscure subtle radiographic cues. After applying LightRefine, these residual artifacts are substantially reduced and the inpainted region becomes more consistent with the surrounding anatomy, exhibiting clearer local texture organization and improved continuity of rib contours and vertebral structures near the former device boundary. For SD-v1.5 Inpaint (bottom), the backbone output is more prone to diffusion-specific artifacts, including intensity inconsistencies, "glow" patterns, and mismatched textures within the masked area. LightRefine effectively suppresses these artifacts and better aligns the filled region with adjacent anatomical context, yielding more realistic textures and clearer structural continuity while keeping unmasked regions unchanged.

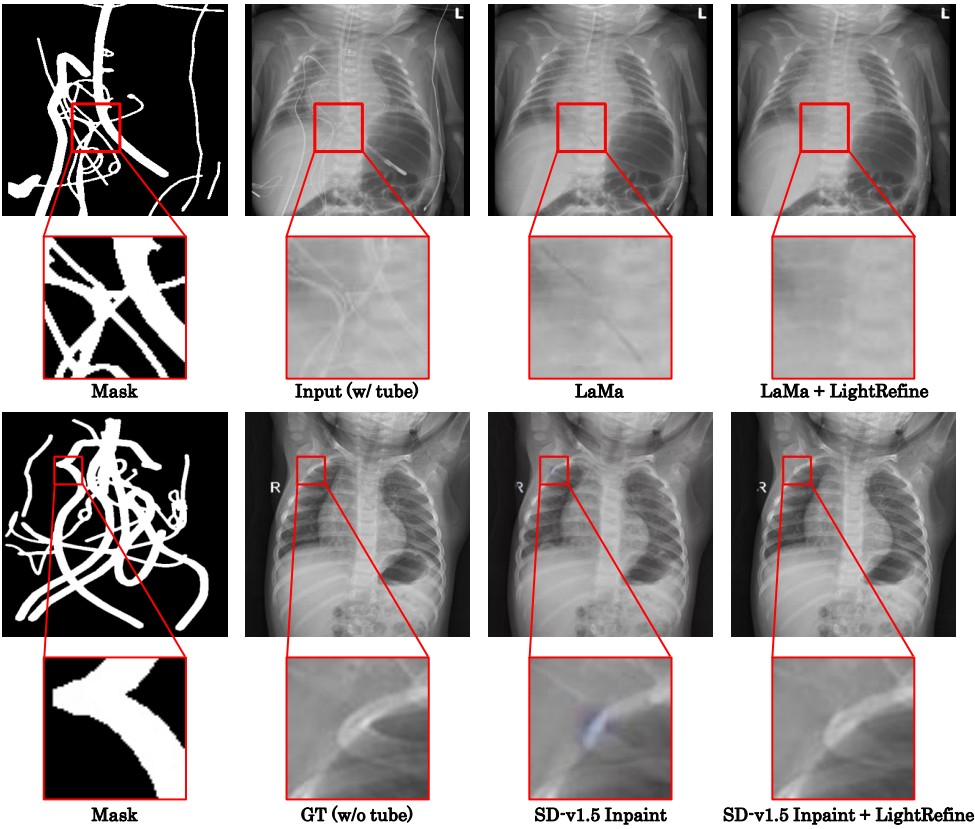

Figure 3: **Qualitative comparisons on pediatric CXRs.** For each example, we show a cropped view (boxed) and a zoomed square ROI. Columns are: **Mask**, **Input (w/ devices)**, **Backbone output**, and **Backbone + LightRefine (ours)**. Top: LaMa. Bottom: SD-v1.5 Inpaint.

### 4.3. Resource efficiency and data-fraction ablation

Table 2 reports LaMa-based results for three settings: backbone-only inference, end-to-end supervised fine-tuning (**LaMa-SFT**), and **LaMa + LightRefine**, together with resource indicators (trainable parameters and peak GPU memory) and performance on VinDr-PCXR test and CHD-CXR. All trainable configurations in this table are evaluated under the same fixed wall-clock training budget (12 hours), enabling a controlled comparison of accuracy–efficiency trade-offs. Within this unified setup, we analyze (i) the resource efficiency of adapting LaMa and (ii) the data efficiency of the refiner through a data-fraction ablation.

**Efficiency comparison under a fixed training budget.** Under the 12-hour budget, **LaMa + LightRefine** attains competitive accuracy while substantially reducing training cost relative to **LaMa-SFT**. Specifically, it trains 24.45M parameters compared with 50.98M for LaMa-SFT and reduces peak GPU memory to approximately 0.45× (32,364 MiB compared with 71,978 MiB), despite using a larger batch size (32 rather than 15).

Table 2: **Efficiency comparison and data-fraction ablation on LaMa under a fixed time budget.** We compare LaMa backbone-only, end-to-end fine-tuning (LaMa-SFT), LaMa + LightRefine, and a parameter-efficient fine-tuning baseline (LaMa+DoRA) under a fixed wall-clock training budget (12 hours). For LaMa + LightRefine, we further study different training data fractions (10%/20%/50%/100% of the VinDr-PCXR training portion after holding out validation). All trainable settings use a shared learning rate of $1 \times 10^{-4}$. We also report training batch size, trainable parameters, and peak GPU memory. **Abbrev.:** Frac.=data fraction, BS=batch size, Params=trainable parameters, Mem=peak GPU memory.

| Method | Frac. | BS | Params (M) | Mem (MiB) | VinDr PSNR↑ | VinDr SSIM↑ | CHD PSNR↑ | CHD SSIM↑ |
|---|---|---|---|---|---|---|---|---|
| LaMa (backbone-only) | – | – | – | – | 41.23 | 0.9782 | 40.39 | 0.9746 |
| LaMa-SFT | 100% | 15 | 50.98 | 71978 | 43.07 | 0.9825 | 41.87 | **0.9786** |
| LaMa+LightRefine (*ours*) | 10% | 32 | 24.45 | 32364 | 42.37 | 0.9800 | 41.28 | 0.9755 |
| | 20% | 32 | 24.45 | 32364 | 42.64 | 0.9807 | 41.46 | 0.9764 |
| | 50% | 32 | 24.45 | 32364 | 42.80 | 0.9811 | 41.64 | 0.9768 |
| | 100% | 32 | 24.45 | 32364 | **43.41** | **0.9828** | **42.01** | 0.9782 |
| LaMa+DoRA | 100% | 15 | 2.54 | 60668 | 40.85 | 0.9768 | 39.62 | 0.9721 |

With the 100% training setting in our protocol, **LaMa + LightRefine** achieves higher PSNR than **LaMa-SFT** and comparable SSIM on both VinDr-PCXR and the external CHD-CXR test set, indicating the ability of **LightRefine** to deliver high-quality device suppression without costly end-to-end backbone updates.

**Data-fraction ablation for refiner training.** Across all training fractions, **LaMa + LightRefine** consistently outperforms backbone-only inference on both VinDr-PCXR and CHD-CXR, with accuracy improving as more data are available. Notably, the 10% setting already yields substantial gains and remains close to full supervised fine-tuning, within 0.70 PSNR on VinDr-PCXR and 0.59 PSNR on CHD-CXR of **LaMa-SFT**. This compelling performance under scarce supervision highlights the data efficiency of LightRefine and its ability to transfer anatomy-consistent corrections to external pediatric CXRs.

**Additional PEFT baseline.** We additionally explored a parameter-efficient fine-tuning baseline for LaMa using DoRA-style (Liu et al., 2024) weight decomposition with low-rank adaptation applied to all `Conv2d` layers. This setting introduces only 2.54M trainable parameters but still requires backpropagation through the full generator backbone, resulting in a relatively high peak memory footprint (60,668 MiB). Under the fixed training budget, we observed slower convergence and weaker reconstruction quality compared with the frozen pretrained backbone in our experiments.

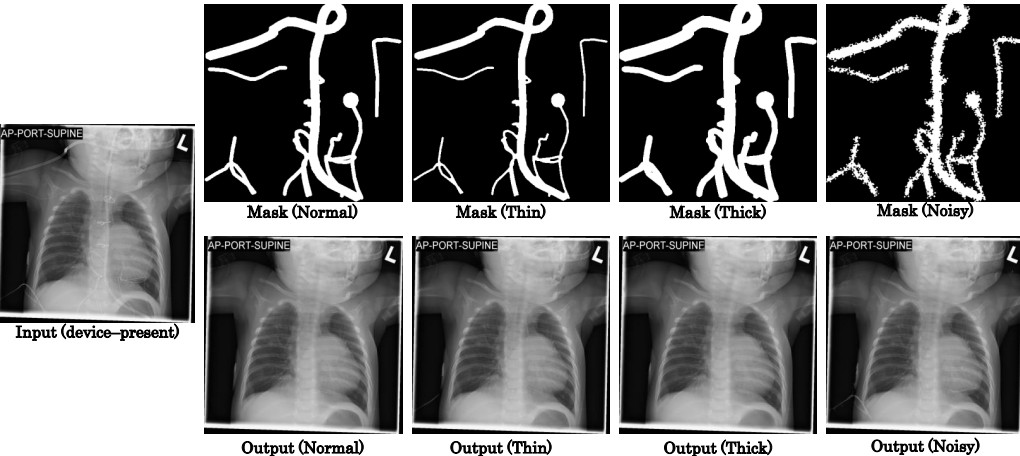

Figure 4: **Robustness to imperfect device masks.** Top: device-present input. Bottom: robustness under four mask conditions, where each column shows the **mask** (row 2) and the corresponding **refined output** (row 3): **Normal** (manual annotation), **Thin** (under-segmentation), **Thick** (over-segmentation), and **Boundary-noisy** (block-wise contour irregularities).

## 4.4. Robustness to imperfect device masks

Fig. 4 illustrates the robustness of **LightRefine** (MAT-based) under four representative imperfect device mask conditions, including a normal manual mask, thin under-segmented masks, thick over-segmented masks, and masks with boundary noise. These settings reflect common inaccuracies encountered in clinical practice when device masks are obtained from manual annotation or automatic segmentation models. Across all conditions, LightRefine produces visually stable and anatomically coherent reconstructions, with refined results remaining consistent under both under- and over-segmentation and without introducing noticeable structural discontinuities or artifact amplification near the edited regions. Even in the presence of noisy mask boundaries, the proposed method effectively mitigates mask-induced artifacts and maintains smooth transitions to surrounding anatomy. Overall, these results indicate that LightRefine is not overly sensitive to moderate mask inaccuracies and can adapt to realistic variations in mask quality, facilitating seamless integration with automatic segmentation pipelines and supporting reliable deployment in real-world pediatric radiography workflows.

## 4.5. Loss ablation study

Table 3 and Fig. 5 jointly present the quantitative and qualitative effects of different loss components on the VinDr-PCXR test set. Residual supervision alone (A1) leads to substantial improvements in PSNR and SSIM, as shown in Table 3, indicating accurate pixel-wise correction within the masked region; however, the visual comparisons in Fig. 5 reveal noticeably smooth and blurred reconstructions. To better capture such perceptual differences beyond distortion-based metrics, we include FID (Heusel et al., 2017) as a complementary

Table 3: **Loss ablation study on VinDr-PCXR test.** We evaluate different combinations of loss terms for the refiner. Metrics include PSNR, SSIM, LPIPS, and FID.

| Setting | Loss terms | | | | VinDr-PCXR test | | | |
|---|---|---|---|---|---|---|---|---|
| | $\mathcal{L}_{res}$ | $\mathcal{L}_{edge}$ | $\mathcal{L}_{tv}$ | $\mathcal{L}_{lpips}$ | PSNR↑ | SSIM↑ | LPIPS↓ | FID↓ |
| LaMa (backbone) | – | – | – | – | 41.23 | 0.9782 | 0.01242 | 4.18 |
| A1 | ✓ | | | | 44.61 | 0.9868 | 0.00967 | 6.89 |
| A2 | ✓ | ✓ | | | 44.35 | 0.9857 | 0.00852 | 5.11 |
| A3 (Full) | ✓ | ✓ | ✓ | ✓ | 43.73 | 0.9837 | 0.00678 | 2.38 |

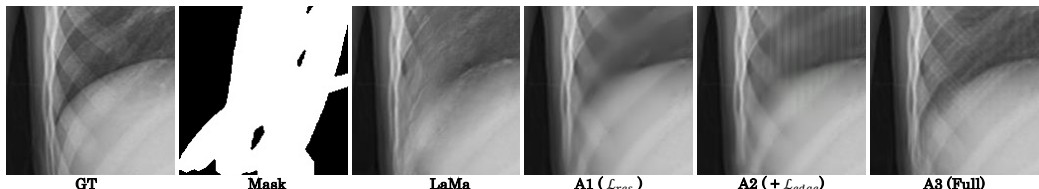

Figure 5: **Zoom-in qualitative loss ablation on a representative case.** We show a cropped ROI (zoom-in) for clearer comparison. From left to right: input image, device mask, LaMa backbone output, and refined results under different loss configurations (A1–A3).

distribution-level indicator alongside LPIPS. Introducing edge consistency constraints (A2) improves local structural continuity and recovers clearer anatomical details, which is reflected by reduced LPIPS and FID values, although minor high-frequency artifacts remain visible in the zoomed regions. The full loss configuration (A3), which further incorporates total variation and masked perceptual supervision, achieves the most balanced performance across complementary metrics. As evidenced by both Table 3 and Fig. 5, A3 produces reconstructions that are visually closest to the LaMa baseline while better preserving fine anatomical details and suppressing artifacts. Although PSNR and SSIM decrease slightly, this trade-off is expected, as perceptual and structural losses promote more realistic image distributions that are not fully captured by pixel-wise fidelity metrics.

## 5. Conclusions

We introduced LightRefine-PCXR, a lightweight and backbone-agnostic framework for medical device suppression in pediatric chest X-rays. By augmenting a frozen pretrained inpainting backbone with a compact anatomy-aware refiner that predicts mask-constrained residuals, the method strictly preserves unmasked regions while improving reconstruction fidelity within device areas. Evaluations on VinDr-PCXR and CHD-CXR demonstrate consistent gains over backbone-only inference with substantially reduced training cost.

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

## Appendix A. Additional Qualitative Results

This appendix provides additional qualitative results to complement the quantitative evaluation in the main paper, including representative comparisons, real clinical examples, and typical failure cases.

**Qualitative Comparison on VinDr-PCXR**    Figure 6 presents qualitative comparisons on the VinDr-PCXR test set, illustrating the effect of the proposed LightRefine module across different inpainting backbones. For each example, we visualize the input mask, the corresponding ground-truth image, and the outputs produced by the backbone models with and without refinement. Regions of interest are highlighted and enlarged to facilitate detailed visual comparison. Overall, LightRefine consistently improves local texture continuity and suppresses backbone-specific artifacts across different methods, while preserving the global anatomical structure of the original predictions.

**Qualitative Results on Real Clinical Data**    Figure 7 shows qualitative results on real pediatric chest radiographs. Each row corresponds to one clinical case containing medical devices, and the columns display the input mask, the original PCXR image, and the corresponding device-suppressed result generated by LaMa with LightRefine. These examples demonstrate the practical applicability of the proposed framework in real clinical scenarios, where device appearances are diverse and often more complex than those observed in synthetic benchmark datasets.

**Failure Cases**    Figure 8 illustrates representative failure cases from both the VinDr-PCXR test set and real clinical data from the BCH dataset. Red arrows indicate regions with visible artifacts, such as residual device structures or hallucinated patterns. We observe that failure cases commonly occur when the backbone output is severely degraded or when the input mask does not fully cover elongated medical devices. In practical deployment, we recommend applying a simple mask dilation step prior to inpainting, as overly thin masks are more likely to miss parts of the device and lead to suboptimal suppression results.

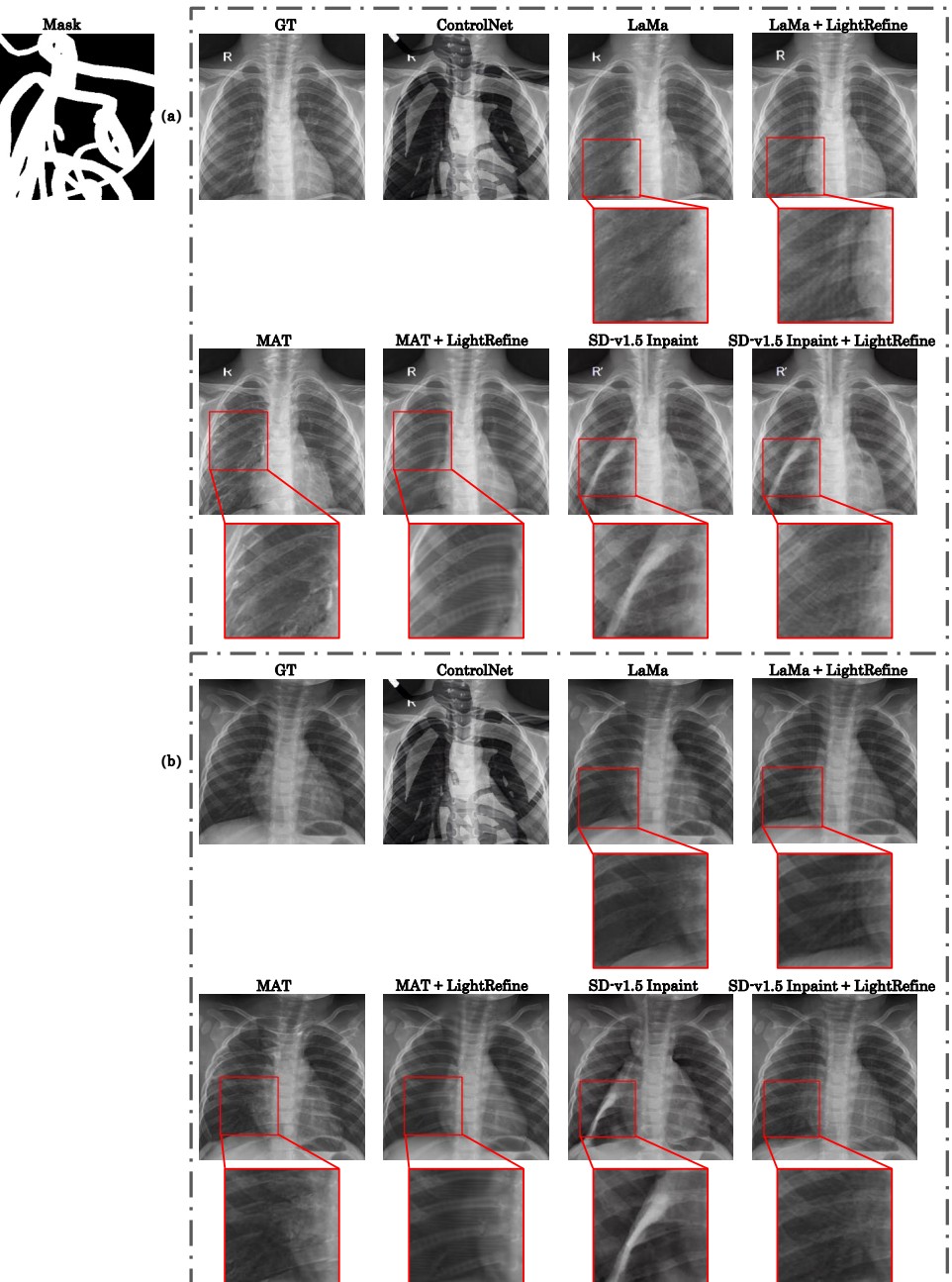

Figure 6: **Qualitative comparison on the VinDr-PCXR test set.** Two representative test cases are shown. The top row presents the input mask, ground truth (GT), and outputs of ControlNet and backbone inpainting methods. The bottom row shows the corresponding LightRefine results applied to the same backbones. Red boxes mark regions of interest (ROIs), with enlarged patches highlighting local details. LightRefine is applied to LaMa, MAT, and SD-v1.5 Inpaint, while ControlNet is shown as a standalone baseline.

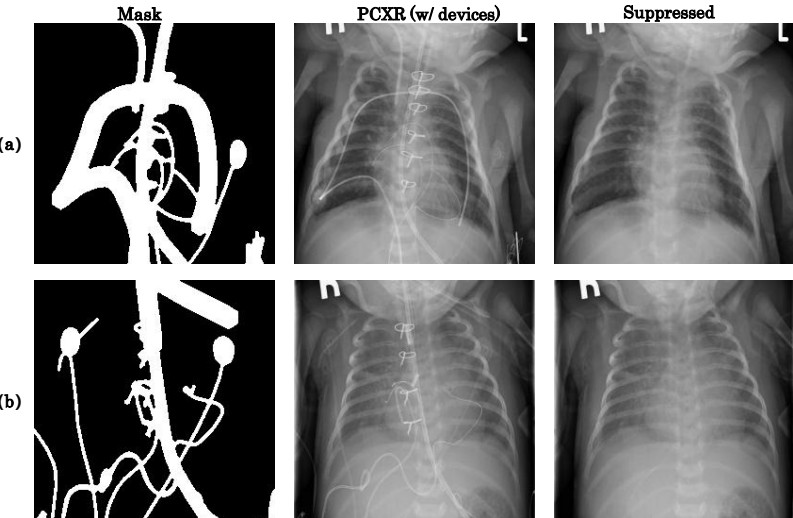

Figure 7: **Qualitative results on real clinical cases.** Each row shows one pediatric chest radiograph with medical devices. From left to right: the input mask, the original PCXR image, and the corresponding device-suppressed result produced by LaMa+LightRefine.

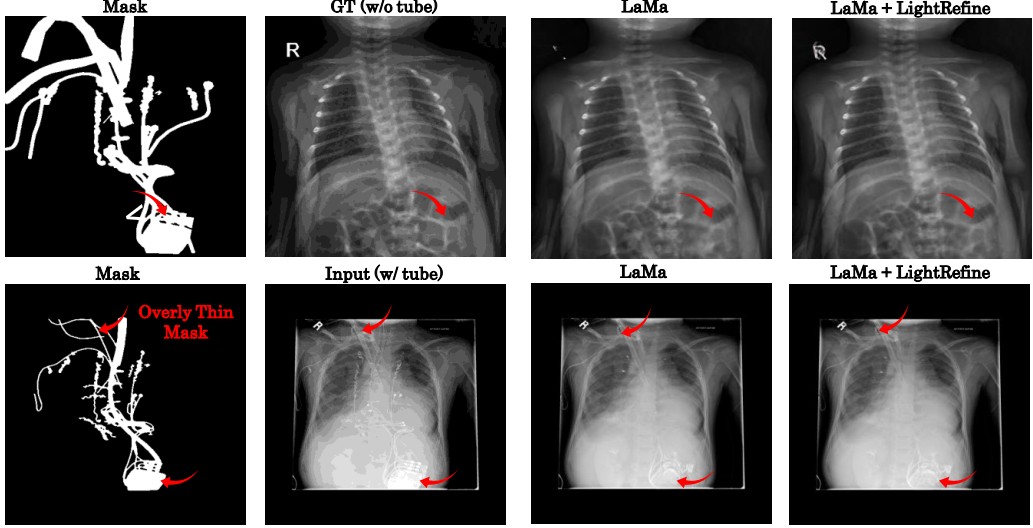

Figure 8: **Failure cases of device suppression. Top:** A representative failure example from the VinDr-PCXR test set. **Bottom:** A failure case from a real clinical scenario. For each example, we show the input mask, the corresponding input or ground-truth image, and the results produced by LaMa and LaMa+LightRefine. Red arrows indicate regions with visible artifacts, such as residual device structures or hallucinated patterns.

Table 4: **Comparison between end-to-end fine-tuning (SFT) and LightRefine under different data fractions.** We report reconstruction performance on VinDr-PCXR test (in-domain) and CHD (external) using PSNR, SSIM, LPIPS, and FID. For each data fraction and dataset, the better result between SFT and LightRefine is highlighted in bold.

| Frac. | Method | VinDr-PCXR test | | | | CHD | | | |
|---|---|---|---|---|---|---|---|---|---|
| | | PSNR↑ | SSIM↑ | LPIPS↓ | FID↓ | PSNR↑ | SSIM↑ | LPIPS↓ | FID↓ |
| 10% | LaMa-SFT | **42.43** | **0.9809** | 0.0096 | 2.93 | **41.39** | **0.9770** | 0.0101 | 4.44 |
| | LightRefine | 42.37 | 0.9800 | **0.0086** | **2.73** | 41.28 | 0.9755 | **0.0096** | **4.33** |
| 20% | LaMa-SFT | 42.45 | **0.9809** | 0.0094 | 2.90 | 41.27 | **0.9769** | 0.0100 | 4.38 |
| | LightRefine | **42.64** | 0.9807 | **0.0082** | **2.63** | **41.46** | 0.9764 | **0.0095** | **4.22** |
| 50% | LaMa-SFT | 42.66 | **0.9813** | 0.0093 | 2.82 | 41.61 | **0.9775** | 0.0098 | 4.20 |
| | LightRefine | **42.80** | 0.9811 | **0.0078** | **2.52** | **41.64** | 0.9768 | **0.0089** | **3.99** |
| 100% | LaMa-SFT | 43.07 | 0.9825 | 0.0084 | 2.63 | 41.87 | **0.9786** | 0.0092 | 3.95 |
| | LightRefine | **43.41** | **0.9828** | **0.0071** | **2.38** | **42.01** | 0.9782 | **0.0084** | **3.79** |

## Appendix B. Additional Data-Fraction Analysis

Table 4 provides a supplementary comparison between end-to-end fine-tuning (SFT) and the proposed LightRefine framework under different training data fractions (10%, 20%, 50%, and 100% of the VinDr-PCXR training set), complementing the main efficiency analysis reported in Table 2. All experiments follow the same training protocol as Table 2, including identical data splits, model backbones, optimization settings, and a fixed wall-clock training budget of 12 hours.

Overall, LightRefine consistently matches or exceeds SFT at moderate to full data regimes (50% and 100%) across both in-domain (VinDr-PCXR) and external (CHD) evaluations. At very small data fractions (10%–20%), performance exhibits mild fluctuations, which we attribute to limited supervision and increased variance from random mask sampling; in this regime, SFT can occasionally achieve slightly higher distortion metrics due to its higher adaptation capacity. As the training data increases, however, the advantage of the proposed refinement strategy becomes more pronounced, yielding more stable improvements, particularly in perceptual quality metrics.

Importantly, these accuracy gains are achieved with substantially lower computational cost. As shown in Table 2, LightRefine requires less than half of the peak GPU memory of SFT and converges significantly faster under the same training budget, since only a compact refinement module is optimized while the backbone remains frozen. These results highlight that LightRefine offers a favorable accuracy–efficiency trade-off for pediatric device suppression, especially in realistic data regimes where both robustness and resource efficiency are critical.

## Appendix C. ControlNet Baseline

Table 5: **Quantitative evaluation of the ControlNet baseline.** All results are computed using hard composition, where pixels outside the masked region are copied from the input image.

| Dataset | PSNR↑ | SSIM↑ | LPIPS↓ | FID↓ |
|---|---|---|---|---|
| VinDr-PCXR (test) | 26.71 | 0.9012 | 0.1460 | 57.95 |
| CHD-CXR | 25.27 | 0.8981 | 0.1529 | 77.02 |

Table 6: **Peak GPU memory usage for SD-v1.5-based methods.** All methods are trained for the same number of epochs (8 epochs) on a single NVIDIA A100 GPU.

| Method | Peak GPU Memory (MiB) |
|---|---|
| SD-v1.5 Inpaint + ControlNet | 37,690 |
| SD-v1.5 Inpaint + LightRefine (ours) | 19,174 |

We additionally evaluate a ControlNet (Zhang et al., 2023) baseline following the publicly released official configuration, which represents a widely used controllable image generation approach that conditions a frozen diffusion backbone on auxiliary spatial inputs. In this setting, the Stable Diffusion backbone is kept frozen and only the ControlNet branch is optimized.

Table 5 reports quantitative results on VinDr-PCXR and CHD-CXR under the same strict inpainting protocol used throughout this work, where pixels outside the masked region are preserved by hard composition. Under this protocol, the ControlNet baseline yields substantially lower PSNR and SSIM and markedly higher LPIPS and FID than inpainting-specific backbones and their refined variants. Qualitative examples in Fig. 6 further show that ControlNet produces visible artifacts and spatial inconsistencies within the masked regions, limiting its suitability for clinical device suppression.

We note that this behavior is largely attributable to a mismatch between the ControlNet formulation and reconstruction-focused medical inpainting. While hard composition avoids penalizing global image changes, ControlNet performs unconstrained generation and does not explicitly enforce pixel-level fidelity within the masked region. As a result, reconstruction errors inside the target area dominate the evaluation, leading to consistently poor performance under medical inpainting metrics.

