# OpenReview forum: "LightRefine-PCXR: A Lightweight Refinement Framework for Efficient Medical Device Suppression in Pediatric Chest X-Rays"
_MIDL.io/2026/Conference — MIDL 2026 Poster_

### Official Review · Reviewer_SDru · 2026-01-10

**Confidence:** 4
**Preliminary Rating:** 3
**Final Rating:** 4

**Summary:**

This paper proposes LightRefine-PCXR, a lightweight two-stage framework for medical device suppression in pediatric chest X-rays. The method freezes a pretrained inpainting backbone to produce a coarse device-removed estimate and introduces a compact refinement network that predicts mask-constrained residual corrections, guaranteeing that unmasked anatomy is preserved exactly.

**Strengths:**

1. The proposed frozen-backbone plus mask-constrained residual refinement strategy significantly reduces training cost and memory usage while guaranteeing that unmasked anatomical regions remain unchanged, addressing both efficiency and safety concerns in clinical deployment.
2. The framework is backbone-agnostic and demonstrates consistent performance improvements across CNN-, transformer-, and diffusion-based inpainting models, as well as robust cross-dataset generalization in low-data pediatric settings.

**Weaknesses:**

1. The evaluation largely relies on synthetic mask-based reconstruction and does not fully capture deployment scenarios with real device-present images and imperfect masks.
2. The experimental validation is limited to image reconstruction metrics, without assessing clinical relevance or impact on downstream tasks.
3. Comparisons to alternative lightweight or anatomy-guided inpainting methods are limited.
4. Cross-dataset experiments follow the same reconstruction protocol as training, providing limited evidence under more realistic domain shifts.

**Detailed Comments:**

1. The paper would benefit from clearer clarification of the mask sampling and pairing strategy. Further details on whether each image is paired with multiple distinct masks during training, and whether the same masks are reused across splits, would improve clarity and reproducibility.

**Justification Of Final Rating:**

I thank the author for the detailed response. Most of my concerns have been addressed. Given the limitation of paired data, I am still questioning the real-world application of such a method. Based on the quality improvement through the revision, I would like to raise my score accordingly.

**Justification Of The Preliminary Rating:**

The method is carefully designed to preserve unmasked anatomy and demonstrates consistent quantitative improvements and strong cross-dataset performance. However, the experimental evaluation relies heavily on synthetic mask-based reconstruction and limited image-level metrics, leaving some uncertainty regarding robustness in real deployment scenarios and clinical impact. Overall, the work offers a solid and practical contribution, though additional validation would be beneficial to fully substantiate its real-world applicability.

**Questions To Address In The Rebuttal:**

Please refer to the weakness and detailed comments.

---

> ### Author Response · Authors · 2026-01-25
> **Response to Reviewer's Feedback**
>
> We appreciate the reviewer’s valuable insights and constructive feedback. Below, we respond to each question and clarify key aspects of our work.
>
> **Q1. Evaluation under real deployment scenarios**
>
> We agree that realistic deployment involves device-present CXRs with imperfect device masks. However, such evaluation is fundamentally constrained by data availability: paired pediatric CXRs with and without devices do not exist in clinical practice, and no public dataset provides such paired references. As a result, direct quantitative evaluation under true deployment conditions is not feasible. We therefore adopt a synthetic mask-based reconstruction protocol, which is standard in image inpainting literature. To better reflect deployment behavior beyond idealized masks, we include a **robustness study with imperfect masks (Section 4.4)**, evaluating under-segmentation, over-segmentation, and boundary noise. Results show that our method remains stable and anatomically coherent under these realistic perturbations.
>
> In addition, we present **qualitative results on real device-present pediatric CXRs** in **Fig. 3 (top row)** and **Appendix A, Fig. 7**, covering representative and real clinical cases. These results directly reflect the intended deployment scenario and demonstrate effective device suppression while preserving surrounding anatomy.
>
> **Q2. Clinical relevance and downstream task evaluation**
>
> We agree that clinical relevance and downstream impact are important considerations. Our primary goal is to establish a **quantitatively reliable and resource-efficient vision-level framework** for pediatric device suppression. We therefore focus on standard reconstruction and perceptual metrics, cross-dataset generalization, and **extensive qualitative visualizations** that directly demonstrate suppression performance.
>
> Specifically, we include multiple visual comparisons on both synthetic and real device-present pediatric CXRs (e.g., Fig. 3, Fig. 4, and Appendix A), illustrating effective device removal with preserved surrounding anatomy. Assessing diagnostic performance or reader studies requires close clinician collaboration and task-specific annotations. We plan to pursue these studies with pediatric radiologists beyond the scope of this submission.
>
> **Q3. Alternative lightweight and anatomy-guided methods**
>
> We note that medical device (tube/line) suppression in pediatric CXRs remains underexplored, and most existing medical inpainting methods focus on full-image reconstruction rather than targeted device removal with strict anatomical preservation.
>
> To contextualize our approach, we include comparisons with representative lightweight and conditioning-based methods from the general vision literature. **DoRA (PEFT)**, included in **Table 2**, is parameter-efficient but still requires backpropagation through the full backbone and underperforms under the same fixed training budget. **ControlNet**, evaluated in **Appendix C**, performs poorly under strict medical inpainting metrics due to unconstrained global generation and lack of pixel-level fidelity within the masked region.
>
> Most anatomy-guided inpainting methods rely on explicit anatomical priors or modality-specific designs and are therefore not directly comparable to our **backbone-agnostic, mask-constrained refinement framework**, which is designed as a general plug-in across heterogeneous inpainting backbones.
>
> **Q4. On the cross-dataset evaluation protocol**
>
> Our cross-dataset experiments involve meaningful domain shifts. The device mask pool used for testing is fully disjoint from the training pool, and CHD-CXR images are never used during training. All models are trained exclusively on VinDr-PCXR, and evaluation on CHD-CXR reflects generalization across both image domains and device mask distributions.
>
> We agree that evaluation on real device-present CXRs would better reflect deployment. However, without paired device-free references, quantitative metrics such as PSNR or SSIM cannot be computed. We therefore adopt the synthetic protocol for fair quantitative comparison, complemented by **qualitative results on real device-present CXRs** (Fig. 3, Appendix A, Fig. 7), which directly demonstrate practical deployment behavior and robustness under realistic domain shifts.
>
> **Q5. Clarification of mask sampling and image–mask pairing strategy**
>
> We have revised **Section 3.1** to clarify the mask sampling and pairing strategy. The device mask pool is split into disjoint training, validation, and test subsets that strictly mirror the image-level splits to prevent data leakage.
> During training, image–mask pairing is stochastic at the iteration level: at each iteration, a device mask is randomly sampled from the training pool and applied to the input image, so the same CXR is paired with different masks across iterations. During validation and testing, image–mask pairs are fixed for reproducibility and shared across all evaluated methods.

---

### Official Review · Reviewer_SzsN · 2026-01-10

**Confidence:** 3
**Preliminary Rating:** 3
**Final Rating:** 4

**Summary:**

This paper proposed a light-weight framework for refining the Medical Device Suppression in Pediatric Chest X-Rays. Specifically, to implement it, a two-stage (non-trainable and trainable) image inpainting training procedure is presented. A compound loss is proposed for complementary regularizers. Overall, the organization, quality of the paper is good.

**Strengths:**

1. Readability. A clear Figure 1 and detailed derivation of the loss components.

2. Efficiency comparison and data-fraction ablation is provided, which is very practical in the medical/clinical data and resource in-sufficient setting.

3. The framework is tested on different types of backbones including CNN-, transformer-, and diffusion-based models

**Weaknesses:**

The main weakness coming from improving the clearness and results analysis of the paper.

1. Examples of failure cases. It is better to provide examples of images and predictions that are consistently failed in all baselines.

2. This is following the previous point and Table 1. Since this is a two-stage, which the refinement is based on the initial results from stage 1. It is interesting to see how significant is the result of stage-1 impacts on the final result? Are the failure cases coming from the bad coars outcome (y0) ?

3. For Figure 1 and Section 2.4 : Would better mention clearly what the x represent. Are the x corresponds to ground-truth?  (which is the x-ray not containing any device). If so, add it into the Figure 1 caption for better interpretation.

"During training, given the ground-truth device-free image x"

**Detailed Comments:**

The paper proposes a efficient framework for Efficient Medical Device Suppression in Pediatric Chest X-Rays. The organization and written quality is good. Main concerns come from improving the clearness of the methodology, results analysis.

**Justification Of Final Rating:**

Thanks for providing more details on the methods, examples of failure cases, and ablation studies. After revision, I changed the rating from boarder line (3) to weak accept. After revision, the work can be useful for the community.

**Justification Of The Preliminary Rating:**

The paper proposes a efficient framework for Efficient Medical Device Suppression in Pediatric Chest X-Rays. The organization and written quality is good. Main concerns come from improving the clearness of the methodology, results analysis and ablation study for publication quality.

**Questions To Address In The Rebuttal:**

1. Examples of failure cases. It is better to provide examples of images and predictions that are consistently failed in all baselines.

2. This is following the previous point and Table 1. Since this is a two-stage, which the refinement is based on the initial results from stage 1. It is interesting to see how significant is the result of stage-1 impacts on the final result? Are the failure cases coming from the bad coars outcome (y0) ?

3. For Figure 1 and Section 2.4 : Would better mention clearly what the x represent. Are the x corresponds to ground-truth?  (which is the x-ray not containing any device). If so, add it into the Figure 1 caption for better interpretation.

"During training, given the ground-truth device-free image x" (this is the description from Section 2.4)

4. Ablation study on the contribution of Loss components.

5. In Table 2, explain why data fraction experiments (10%, 20%, ...) are not used in SFT? Were they not converged during training, etc ?

---

> ### Author Response · Authors · 2026-01-25
> **Response to Reviewer's Feedback**
>
> We appreciate the reviewer’s valuable insights and constructive feedback. Below, we provide detailed responses to each question, addressing the raised concerns and clarifying key aspects of our work. In the Supporting Material of this rebuttal, we have also included an updated version of the manuscript, in which the relevant clarifications and additions have been incorporated.
>
> **Q1 & Q2. Examples of failure cases**
>
> We thank the reviewer for these related questions. In the revised manuscript, we address both points jointly through additional qualitative analysis.
>
> First, we include **examples of failure cases that are consistently challenging across all baselines** in **Appendix A, Figure 6**. In our two-stage framework, the Stage-1 output y(0) is exactly the prediction produced by the inpainting backbone, i.e., the **LaMa / MAT / SD-v1.5 Inpaint outputs shown in the figure**. As illustrated, these cases are difficult for all backbone methods. Nevertheless, the refined results show **substantial improvements over the corresponding Stage-1 outputs**, producing reconstructions that are visibly closer to the ground truth, which demonstrates that the refiner can effectively correct many backbone-level errors.
>
> Second, we explicitly analyze the remaining failure modes in **Appendix A, Figure 8**. These failures are primarily observed when (i) the input mask is extremely thin or severely incomplete, or (ii) the Stage-1 backbone output is already highly degraded or dominated by strong hallucinations. In such cases, the refinement quality is inevitably affected, as the refiner operates on the backbone prediction and is constrained to the masked region.
>
> **Q3. Definition of x**
>
> Clarified in the revised manuscript. x denotes the ground-truth device-free pediatric chest X-ray.
>
> **Q4. Ablation study on the contribution of Loss component**
>
> We thank the reviewer for this suggestion. In the revised manuscript, we include a dedicated loss ablation study in Section 4.5, where we analyze the contribution of individual loss components using both quantitative metrics and qualitative visualizations. The ablation compares residual supervision alone, the addition of edge consistency, and the full loss configuration. As reported in the updated table, residual-only supervision yields higher pixel-wise metrics, while the full configuration achieves improved LPIPS and FID, indicating better perceptual and structural quality. These numerical trends are further supported by zoom-in qualitative examples, which show clearer anatomical continuity and fewer artifacts with the full loss design. Together, the results demonstrate that our final loss formulation provides a more balanced and anatomically plausible reconstruction rather than optimizing a single metric.
>
> **Q5. Data fraction experiments for SFT in Table 2**
>
> In this work, we consider *resource efficiency* from two complementary perspectives: **(i) computational efficiency** and **(ii) data efficiency**, both of which are reflected in Table 2.
>
> For **computational efficiency**, we compare end-to-end fine-tuning (SFT) with LightRefine under a fixed wall-clock training budget of 12 hours and report practical training costs, including peak GPU memory usage and the number of trainable parameters. Under this controlled setting, LightRefine achieves comparable or better performance than SFT while requiring **substantially fewer trainable parameters and less than half of the peak GPU memory**.
>
> The **data fraction analysis** in Table 2 is intended to evaluate data efficiency, focusing on how effectively a method adapts under varying amounts of training data. This aspect is particularly relevant in pediatric imaging, where labeled data are often limited. As shown in the fraction study, LightRefine maintains strong performance even at small data fractions and scales reliably as more data become available, demonstrating stable adaptation behavior and good data efficiency.
>
> For clarity and conciseness, Table 2 emphasizes the scaling behavior of LightRefine, while the full-data SFT result is used as a representative reference point for the computational efficiency comparison. **To provide a more complete view, we have added the corresponding SFT results at 10%, 20%, and 50% data fractions in Appendix B.** These additional results show that LightRefine consistently matches or exceeds SFT at moderate to full data scales, while retaining clear advantages in computational cost and convergence efficiency.

---

### Official Review · Reviewer_AXLd · 2026-01-13

**Confidence:** 4
**Preliminary Rating:** 3
**Final Rating:** 4

**Summary:**

The authors propose LightRefinePCXR, a lightweight, backbone-agnostic refinement framework for suppressing medical devices in pediatric chest X-rays. The framework contains a two-stage approach, first a frozen pretrained inpainting backbone that can be backbone-agnostic, and a compact anatomy-aware refiner predicting mask-constrained residual corrections. On two chest X-ray public datasets, the authors tested the refinement method on three inpainting backbones, including CNN, Transformer, and Diffusion model, and reported improved anatomical fidelity.

**Strengths:**

The two-stage refinement design is clear and straightforward. The refinement can be used as a backbone-agnostic method supporting CNN, Transformer, and Diffusion models.

The efficiency problem is addressed and the refinement network is lightweighted, showing improvement in not only performance but also time and memory consumption, supporting practical pediatric adaptation under limited data and compute constraints.

**Weaknesses:**

Though the authors claimed that their method doesn't require paired masks and images, they still need to acquire an annotation set for multiple devices. Through their training, they randomly sample a device mask and mask out a device-free CXR. The need for a mask set might still be a limitation. When there is a device on the subject, there might be metal artifacts, and thus a masked-out device-free CXR might not be as realistic as a CXR when devices are actually present.

The refiner design is relatively lacking in technical novelty: a concatenated input sent into a U-Net. Given there are already techniques introducing condition to generative backbone such as ControlNet, the design of this refinement network might need to be further discussed. In addition, the comparison with baselines only included results without and with the proposed refinement, in addition to an SFT method. Additional comparisons with other refinement methods including ControlNet might be needed.

**Detailed Comments:**

An ablation study with imperfect device masks would help with evaluating mask dependency and robustness in real clinical settings. Also, discussing if using automatic device segmentation rather than manual masks would help with deployability.

**Justification Of Final Rating:**

The authors have substantially addressed the main concerns. Given that synthetic device masking might not be fully representing physical effects/artifacts of real devices, I am leaning toward weak acceptance.

**Justification Of The Preliminary Rating:**

The authors propose a unified, backbone-agnostic refinement framework for device removal, with improved performance on various backbones. However, further comparison with more recent refinement methods is missing, and the dependency on device mask might be a limitation. Thus, I am giving my initial borderline rating.

**Questions To Address In The Rebuttal:**

See weaknesses and comments

---

> ### Author Response · Authors · 2026-01-25
> **Response to Reviewer's Feedback**
>
> We appreciate the reviewer’s valuable insights and constructive feedback. Below, we respond to each question and clarify key aspects of our work.
>
> **Q1. Mask supervision and realism of synthetic device masking**
>
> We agree that our approach relies on a set of device masks and that synthetic masking cannot fully capture all aspects of real device-present CXRs, such as metal-induced artifacts.
>
> When we state that our method does not require *paired* data, we refer to the absence of paired device-present and device-free CXRs of the same patient, which are rarely available in clinical practice. Instead, training uses a reusable pool of device masks randomly applied to device-free CXRs. Compared to paired image supervision, such mask annotations are lighter-weight, easier to acquire, and can also be obtained from existing datasets or automatic segmentation models.
>
> While some inpainting or suppression methods for natural images do not require explicit masks (e.g., fully generative or diffusion-based approaches), unconstrained generation may introduce geometric distortions or global appearance shifts that are unacceptable in medical imaging, where fine anatomical details are clinically meaningful. By explicitly localizing the edited region with device masks, our method guarantees that pixels outside the mask remain unchanged via hard compositing, preserving diagnostic content and clinical safety.
>
> Regarding realism, we acknowledge that masking devices on device-free CXRs does not model all physical effects of real devices. Our synthetic protocol is therefore primarily designed to enable controlled and quantitative evaluation, rather than to perfectly simulate all deployment conditions. To partially bridge this gap, we present qualitative results on real device-present pediatric CXRs (Fig. 3, top row; Fig. 7) and a robustness study with imperfect masks (Section 4.4), demonstrating stable performance under realistic deviations. More faithful modeling of device-related artifacts and tighter integration with real device-present data remain important future directions.
>
> **Q2. Additional refinement and conditioning baselines**
>
> Our experiments show that the refiner is highly effective for the **targeted task of medical device (e.g., tube and line) suppression**, consistently improving reconstruction quality across multiple backbones and datasets, and in several cases outperforming more complex adaptation or conditioning approaches. Pediatric device suppression is a **clinically meaningful yet underexplored task**, and our method reliably improves anatomical clarity while reducing device-related artifacts.
>
> Methodologically, refiner-based approaches that explicitly enforce **mask-constrained residual corrections** remain largely unexplored in medical image inpainting. Existing methods mainly focus on conditioning or fine-tuning large generative backbones, whereas our work introduces a lightweight refinement paradigm tailored to medical imaging, with explicit guarantees on anatomical preservation and deployment reliability.
>
> Regarding additional baselines, we include the following comparisons in the revised manuscript:
>
> 1. **ControlNet**: Evaluated using the publicly released official implementation, with results reported in **Appendix C** (Tables 5–6 and Fig. 6). Under the same strict medical inpainting protocol, ControlNet shows inferior reconstruction fidelity, attributed to unconstrained global generation and the lack of explicit pixel-level enforcement within the masked region.
> 2. **DoRA (PEFT)**: A low-rank adaptation baseline is included in **Table 2**. While parameter-efficient, it still requires backpropagation through the full backbone and underperforms under the same fixed training budget.
>
> **Q3. Mask dependency and robustness under imperfect device masks**
>
> As suggested, we conducted an additional ablation study evaluating robustness under imperfect device masks, which are common in clinical settings, with results included in **Section 4.4**. We perturb device masks at test time to simulate under-segmentation, over-segmentation, and boundary noise. LightRefine-PCXR maintains stable and anatomically coherent reconstructions across all variants, with modifications strictly confined to masked regions by design, indicating robustness to realistic mask inaccuracies.
>
> **Q4. Use of automatic device segmentation for practical deployment**
>
> In this work, our focus is on isolating the device suppression problem itself, for which manually annotated masks provide a reliable and controlled evaluation setting. As shown in **Section 4.4**, our method remains stable under mask imperfections typical of automatic segmentation models, including boundary imprecision and under- and over-segmentation. These results suggest that integrating automatic device segmentation is feasible without fundamentally affecting the proposed framework, and we view such end-to-end integration as a natural direction for future work.

---

### Author Rebuttal · Authors · 2026-01-25

**Rebuttal:**

We sincerely appreciate the valuable feedback from the reviewers, as well as the efforts of the program chairs and area chairs in facilitating this review process. Based on the insightful comments provided, we have carefully revised the manuscript to address the concerns raised. All changes are highlighted in the revised manuscript for clarity.

**Supporting Material:**

/attachment/cc8aa3a92709cc3da31959ea98d03a74186a0f05.pdf

---

### Meta-Review · Area_Chair_BNDJ · 2026-02-07

**Recommendation:** Accept (Poster)
**Confidence:** 4

**Metareview:**

This paper presents LightRefine-PCXR, a lightweight, plug-in refinement framework designed to improve the quality of medical device (e.g., tubes, lines) suppression in pediatric chest X-rays. The core idea is a two-stage, backbone-agnostic approach: a frozen, pre-trained inpainting model (e.g., LaMa, MAT, Stable Diffusion) generates a coarse "cleaned" image, and a small, trainable "refiner" network then predicts residual corrections strictly within the device mask region. This design explicitly preserves all unmasked anatomy, reduces computational cost versus full fine-tuning, and demonstrates consistent improvements across multiple inpainting backbones and datasets.

After review and author rebuttal, the consensus among reviewers shifted from Borderline to a clear Weak Accept. The authors have adequately addressed the primary concerns through clarifications in the manuscript and the addition of new experiments (robustness to imperfect masks, comparison with ControlNet, loss ablation). The work is deemed a solid, practical contribution with clear strengths in efficiency and generalizability. For these reasons, I recommend this paper be accepted.

---

### Decision · Program_Chairs · 2026-02-13

Accept (Poster)